# Deep Discriminative to Kernel Density Graph for In- and Out-of-distribution Calibrated Inference

## Abstract

Deep discriminative approaches like random forests and deep neural networks have recently found applications in many important real-world scenarios. However, deploying these learning algorithms in safety-critical applications raises concerns, particularly when it comes to ensuring confidence calibration for both in-distribution and out-of-distribution data points. Many popular methods for in-distribution (ID) calibration, such as isotonic and Platt's sigmoidal regression, exhibit excellent ID calibration performance. However, these methods are not calibrated for the entire feature space, leading to overconfidence in the case of out-of-distribution (OOD) samples. On the other end of the spectrum, existing out-of-distribution (OOD) calibration methods generally exhibit poor in-distribution (ID) calibration. In this paper, we address ID and OOD calibration problems jointly. We leveraged the fact that deep models, including both random forests and deep-nets, learn internal representations which are unions of polytopes with affine activation functions to conceptualize them both as partitioning rules of the feature space. We replace the affine function in each polytope populated by the training data with a Gaussian kernel. Our experiments on both tabular and vision benchmarks show that the proposed approaches obtain well-calibrated posteriors while mostly preserving or improving the classification accuracy of the original algorithm for ID region, and extrapolate beyond the training data to handle OOD inputs appropriately.

## 1 Introduction

Machine learning methods, specially deep neural networks and random forests have shown excellent performance in many real-world tasks, including drug discovery, autonomous driving and clinical surgery [1–3]. However, calibrating confidence over the whole feature space for these approaches remains a key challenge in the field [4]. Calibrated confidence within the training or in-distribution (ID) region as well as in the out-of-distribution (OOD) region is crucial for safety critical applications like autonomous driving and computer-assisted surgery, where any aberrant reading should be detected and taken care of immediately [4, 5].

The approaches to calibrate OOD confidence for learning algorithms described in the literature can be roughly divided into two groups: discriminative and generative. Intuitively, the easiest solution for OOD confidence calibration is to learn a function that gives higher scores for in-distribution samples and lower scores for OOD samples [6]. The discriminative approaches try to either modify the loss function [7–9] or train the network exhaustively on OOD datasets to calibrate on OOD samples [10, 4]. Recently, Hein et al. [4] showed RELU networks produce arbitrarily high confidence as the inference point moves far away from the training data. Therefore, calibrating RELU networks for the whole OOD region is not possible without fundamentally changing the network architecture. As a result, all of the aforementioned algorithms are unable to provide any guarantee about the performance of the network throughout the whole feature space. The other group tries to learn generative models for the

in-distribution as well as the out-of-distribution samples. The general idea is to do likelihood ratio test for a particular sample using the generative models [11], or threshold the ID likelihoods to detect OOD samples. However, it is not obvious how to control likelihoods far away from the training data for powerful generative models like variational autoencoders (VAEs) [12] and generative adversarial networks (GAN) [13]. Moreover, Nalisnick et al. [14] and Hendrycks et al. [10] showed VAEs and GANs can also yield overconfident likelihoods far away from the training data.

The algorithms described so far are concerned with OOD confidence calibration for deep-nets only. However, we show that other approaches which partition the feature space, for example random forest, can also suffer from poor confidence calibration both in the ID and the OOD regions. Moreover, the algorithms described above are concerned about the confidence in the OOD region only and do not address the confidence calibration within the ID region at all. This issue is addressed separately in a different group of literature [15–20]. Instead, we consider both calibration problems jointly and propose an approach that achieves good calibration throughout the whole feature space.

In this paper, we conceptualize both random forest and ReLU networks as partitioning rules with an affine activation over each polytope. We consider replacing the affine functions learned over the polytopes with Gaussian kernels. We propose two novel kernel density estimation techniques named *Kernel Density Forest* (KDF) and *Kernel Density Network* (KDN). Our proposed approach completely excludes the need for training on OOD examples for the model (unsupervised OOD calibration). We conduct several simulation and real data studies that show both KDF and KDN are well-calibrated for OOD samples while they maintain good performance in the ID region.

## 2 Related Works and Our Contributions

There are a number of approaches in the literature which attempt to learn a generative model and control the likelihoods far away from the training data. For example, Ren et al. [11] employed likelihood ratio test for detecting OOD samples. Wan et al. [8] modified the training loss so that the downstream projected features follow a Gaussian distribution. However, there is no guarantee of performance for OOD detection for the above methods. To the best of our knowledge, apart from us, only Meinke et al. [5] has proposed an approach to guarantee asymptotic performance for OOD detection. Compared to the aforementioned methods, our approach differs in several ways:

- We address the confidence calibration problem for both ReLU-nets and random forests.
- We address ID and OOD calibration problem as a continuum.
- We provide an algorithm for OOD confidence calibration for both tabular and vision datatsets whereas most of the existing methods are tailor-made for vision problems.
- We propose an unsupervised post-hoc OOD calibration approach.

## 3 Technical Background

### 3.1 Setting

Consider a supervised learning problem with independent and identically distributed training samples $\{(\mathbf{x}_i, y_i)\}_{i=1}^n$ such that $(\mathbf{X}, Y) \sim P_{X,Y}$, where $\mathbf{X} \sim P_X$ is a $\mathcal{X} \subseteq \mathbb{R}^D$ valued input and $Y \sim P_Y$ is a $\mathcal{Y} = \{1, \cdots, K\}$ valued class label. Let $\mathcal{S}$ be the high density region of the marginal, $P_X$, thus $\mathcal{S} \subsetneq \mathcal{X}$. Here the goal is to learn a confidence score, $\mathbf{g} : \mathbb{R}^D \to [0, 1]^K$, $\mathbf{g}(\mathbf{x}) = [g_1(\mathbf{x}), g_2(\mathbf{x}), \ldots, g_K(\mathbf{x})]$ such that,

$$g_y(\mathbf{x}) = \begin{cases} P_{Y|X}(y|\mathbf{x}), & \text{if } \mathbf{x} \in \mathcal{S} \\ P_Y(y), & \text{if } \mathbf{x} \notin \mathcal{S} \end{cases}, \quad \forall y \in \mathcal{Y} \tag{1}$$

where $P_{Y|X}(y|\mathbf{x})$ is the posterior probability for class $y$ given by the Bayes formula:

$$P_{Y|X}(y|\mathbf{x}) = \frac{P_{X|Y}(\mathbf{x}|y)P_Y(y)}{\sum_{k=1}^K P_{X|Y}(\mathbf{x}|k)P_Y(k)}, \quad \forall y \in \mathcal{Y}. \tag{2}$$

Here $P_{X|Y}(\mathbf{x}|y)$ is the class conditional density which we will refer as $f_y(\mathbf{x})$ hereafter for brevity.

## 3.2  Main Idea

Deep discriminative networks partition the feature space $\mathbb{R}^d$ into a union of $p$ affine polytopes $Q_r$ such that $\bigcup_{r=1}^p Q_r = \mathbb{R}^d$, and learn an affine function over each polytope [4, 21]. Mathematically, the unnormalized class-conditional density for the label $y$ estimated by these deep discriminative models at a particular point $\mathbf{x}$ can be expressed as:

$$\hat{f}_y(\mathbf{x}) = \sum_{r=1}^p (\mathbf{a}_r^\top \mathbf{x} + b_r)\mathbb{1}(\mathbf{x} \in Q_r). \tag{3}$$

For example, in the case of a decision tree, $\mathbf{a}_r = \mathbf{0}$, i.e., decision tree assumes uniform distribution for the class-conditional densities over the leaf nodes. Among these polytopes, the ones that lie on the boundary of the training data extend to the whole feature space and hence encompass all the OOD samples. Since the posterior probability for a class is determined by the affine activation over each of these polytopes, the algorithms tend to be overconfident when making predictions on the OOD inputs. Moreover, there exist some polytopes that are not populated with training data. These unpopulated polytopes serve to interpolate between the training sample points. If we replace the affine activation function of the populated polytopes with Gaussian kernels and prune the unpopulated ones, the tail of the kernel will help interpolate between the training sample points while assigning lower likelihood to the low density or unpopulated polytope regions of the feature space. This results in better confidence calibration for the proposed modified approach.

## 3.3  Proposed Approach

We will call the above discriminative approaches as the 'parent approach' hereafter. Consider the collection of polytope indices $\mathcal{P}$ from the parent approach which are populated by the training data. We replace the affine functions over the populated polytopes with Gaussian kernels $\mathcal{G}(\cdot; \hat{\mu}_r, \hat{\Sigma}_r)$. For a particular inference point $\mathbf{x}$, we consider the Gaussian kernel with the minimum distance from the center of the kernel to the corresponding point:

$$r_\mathbf{x}^* = \operatorname*{argmin}_r \|\mu_r - \mathbf{x}\|, \tag{4}$$

where $\|\cdot\|$ denotes a distance. As we will show later, the type of distance metric considered in Equation 4 highly impacts the performance of the proposed model. In short, we modify Equation 3 from the parent RELU-net or random forest to estimate the class-conditional density (unnormalized):

$$\tilde{f}_y(\mathbf{x}) = \frac{1}{n_y} \sum_{r \in \mathcal{P}} n_{ry}\mathcal{G}(\mathbf{x}; \mu_r, \Sigma_r)\mathbb{1}(r = r_\mathbf{x}^*), \tag{5}$$

where $n_y$ is the total number of samples with label $y$ and $n_{ry}$ is the number of samples from class $y$ that end up in polytope $Q_r$. We add a small constant to the class conditional density $\tilde{f}_y$:

$$\hat{f}_y(\mathbf{x}) = \tilde{f}_y(\mathbf{x}) + \frac{b}{\log(n)}. \tag{6}$$

Note that in Equation 6, $\frac{b}{\log(n)} \to 0$ as the total training points, $n \to \infty$. The intuition behind the added constant will be clarified further later in Proposition 2. The confidence score $\hat{g}_y(\mathbf{x})$ for class $y$ given a test point $\mathbf{x}$ is estimated using the Bayes rule as:

$$\hat{g}_y(\mathbf{x}) = \frac{\hat{f}_y(\mathbf{x})\hat{P}_Y(y)}{\sum_{k=1}^K \hat{f}_k(\mathbf{x})\hat{P}_Y(k)}, \tag{7}$$

where $\hat{P}_Y(y)$ is the empirical prior probability of class $y$ estimated from the training data. We estimate the class for a particular inference point $\mathbf{x}$ as:

$$\hat{y} = \operatorname*{argmax}_{y \in \mathcal{Y}} \hat{g}_y(\mathbf{x}). \tag{8}$$

## 4  Model Parameter Estimation

### 4.1  Gaussian Kernel Parameter Estimation

We fit Gaussian kernel parameters to the samples that end up in the $r$-th polytope. We set the kernel center along the $d$-th dimension:

$$\hat{\mu}_r^d = \frac{1}{n_r} \sum_{i=1}^{n} x_i^d \mathbb{1}(\mathbf{x}_i \in Q_r), \tag{9}$$

where $x_i^d$ is the value of $\mathbf{x}_i$ along the $d$-th dimension. We set the kernel variance along the $d$-th dimension:

$$(\hat{\sigma}_r^d)^2 = \frac{1}{n_r} \{ \sum_{i=1}^{n} \mathbb{1}(\mathbf{x}_i \in Q_r)(x_i^d - \hat{\mu}_r^d)^2 + \lambda \}, \tag{10}$$

where $\lambda$ is a small constant that prevents $\hat{\sigma}_r^d$ from being $0$. We constrain our estimated Gaussian kernels to have diagonal covariance.

### 4.2  Sample Size Ratio Estimation

For a high dimensional dataset with low training sample size, the polytopes are sparsely populated with training samples. For improving the estimate of the ratio $\frac{n_{ry}}{n_y}$ in Equation 5, we incorporate the samples from other polytopes $Q_s$ based on the similarity $w_{rs}$ between $Q_r$ and $Q_s$ as:

$$\frac{\hat{n}_{ry}}{\hat{n}_y} = \frac{\sum_{s \in \mathcal{P}} \sum_{i=1}^{n} w_{rs} \mathbb{1}(\mathbf{x}_i \in Q_s) \mathbb{1}(y_i = y)}{\sum_{r \in \mathcal{P}} \sum_{s \in \mathcal{P}} \sum_{i=1}^{n} w_{rs} \mathbb{1}(\mathbf{x}_i \in Q_s) \mathbb{1}(y_i = y)}. \tag{11}$$

As $n \to \infty$, the estimated weights $w_{rs}$ should satisfy the condition:

$$w_{rs} \to \begin{cases} 0, & \text{if } Q_r \neq Q_s \\ 1, & \text{if } Q_r = Q_s. \end{cases} \tag{12}$$

For simplicity, we will describe the estimation procedure for $w_{rs}$ in the next sections. Note that if we satisfy Condition 12, then we have $\frac{\hat{n}_{ry}}{\hat{n}_y} \to \frac{n_{ry}}{n_y}$ as $n \to \infty$. Therefore, we modify Equation 5 as:

$$\hat{f}_y(\mathbf{x}) = \frac{1}{\hat{n}_y} \sum_{r \in \mathcal{P}} \hat{n}_{ry} \mathcal{G}(\mathbf{x}; \hat{\mu}_r, \hat{\Sigma}_r) \mathbb{1}(r = \hat{r}_\mathbf{x}^*), \tag{13}$$

where $\hat{r}_\mathbf{x}^* = \operatorname{argmin}_r \|\hat{\mu}_r - \mathbf{x}\|$. Now we use $\hat{f}_y(\mathbf{x})$ estimated using (13) in Equation (6), (7) and (8), respectively. Below, we describe how we estimate $w_{rs}$ for KDF and KDN .

### 4.3  Forest Kernel

Consider $T$ number of decision trees in a random forest trained on $n$ $iid$ training samples $\{(\mathbf{x}_i, y_i)\}_{i=1}^{n}$. Each tree $t$ partitions the feature space into $p_t$ polytopes resulting in a set of polytopes: $\{\{Q_{t,r}\}_{r=1}^{p_t}\}_{t=1}^{T}$. The intersection of these polytopes gives a new set of polytopes $\{Q_r\}_{r=1}^{p}$ for the forest. For any two points $\mathbf{x} \in Q_r$ and $\mathbf{x}' \in Q_s$, we define the kernel $\mathcal{K}(r, s)$ as:

$$\mathcal{K}(r, s) = \frac{t_{rs}}{T}, \tag{14}$$

where $t_{rs}$ is the total number of trees, $\mathbf{x}$ and $\mathbf{x}'$ end up in the same leaf node. Here, $0 \leq \mathcal{K}(r, s) \leq 1$.

*If the two samples end up in the same leaf in all the trees, i.e., $\mathcal{K}(r, s) = 1$, they belong to the same polytope, i.e. $r = s$. In short, $\mathcal{K}(r, s)$ is the fraction of total trees where the two samples follow the same path from the root to a leaf node.* We exponentiate $\mathcal{K}(r, s)$ so that Condition 12 is satisfied:

$$w_{rs} = \mathcal{K}(r, s)^{k \log n}. \tag{15}$$

We choose $k$ using grid search on a hold-out dataset.

## 4.4 Network Kernel

Consider a fully connected $L$ layer ReLU-net trained on $n$ $iid$ training samples $\{(\mathbf{x}_i, y_i)\}_{i=1}^n$. We have the set of all nodes denoted by $\mathcal{N}_l$ at a particular layer $l$. We can randomly pick a node $n_l \in \mathcal{N}_l$ at each layer $l$, and construct a sequence of nodes starting at the input layer and ending at the output layer which we call an **activation path**: $m = \{n_l \in \mathcal{N}_l\}_{l=1}^L$. Note that there are $N = \Pi_{i=1}^L |\mathcal{N}_l|$ possible activation paths for a sample in the ReLU-net. We index each path by a unique identifier number $z \in \mathbb{N}$ and construct a sequence of activation paths as: $\mathcal{M} = \{m_z\}_{z=1,\cdots,N}$. Therefore, $\mathcal{M}$ contains all possible activation pathways from the input to the output of the network.

While pushing a training sample $\mathbf{x}_i$ through the network, we define the activation from a ReLU unit at any node as '1' when it has positive output and '0' otherwise. Therefore, the activation indicates on which side of the affine function at each node the sample falls. The activation for all nodes in an activation path $m_z$ for a particular sample creates an **activation mode** $a_z \in \{0, 1\}^L$. If we evaluate the activation mode for all activation paths in $\mathcal{M}$ while pushing a sample through the network, we get a sequence of activation modes: $\mathcal{A}_r = \{a_z^r\}_{z=1}^N$. Here $r$ is the index of the polytope where the sample falls in.

*If the two sequences of activation modes for two different training samples are identical, they belong to the same polytope.* In other words, if $\mathcal{A}_r = \mathcal{A}_s$, then $Q_r = Q_s$. This statement holds because the above samples will lie on the same side of the affine function at each node in different layers of the network. Now, we define the kernel $\mathcal{K}(r, s)$ as:

$$\mathcal{K}(r, s) = \frac{\sum_{z=1}^N \mathbb{1}(a_z^r = a_z^s)}{N}. \tag{16}$$

Note that $0 \leq \mathcal{K}(r, s) \leq 1$. In short, $\mathcal{K}(r, s)$ is the fraction of total activation paths which are identically activated for two samples in two different polytopes $r$ and $s$. We exponentiate the kernel using Equation 15. Pseudocodes outlining the two algorithms are provided in Appendix D.

## 4.5 Geodesic Distance

Consider $\mathcal{P}_n = \{Q_1, Q_2, \cdots, Q_p\}$ as a partition of $\mathbb{R}^d$ given by a random forest or a ReLU-net after being trained on $n$ training samples. We measure distance between two points $\mathbf{x} \in Q_r, \mathbf{x}' \in Q_s$ using the kernel introduced in Equation 14 and Equation 16, and call it 'Geodesic' distance [22]:

$$d(r, s) = -\mathcal{K}(r, s) + \frac{1}{2}(\mathcal{K}(r, r) + \mathcal{K}(s, s)) = 1 - \mathcal{K}(r, s). \tag{17}$$

**Proposition 1.** $(\mathcal{P}_n, d)$ *is a metric space.*

*Proof.* See Appendix A.1 for the proof. $\qquad\square$

We use Geodesic distance to find the nearest polytope to the inference point. As Geodesic distance cannot distinguish between points within the same polytope, it has a resolution similar to the size of the polytope. For discriminating between two points within the same polytope, we fit a Gaussian kernel within the polytope (described above). As $h_n \to 0$, the resolution for Geodesic distance improves. In Section 5, we will empirically show that using Geodesic distance scales better with higher dimension compared to that of Euclidean distance.

Given $n$ training samples $\{(\mathbf{x}_i, y_i)\}_{i=1}^n$, we define the distance of an inference point $\mathbf{x}$ from the training points as: $d_{\mathbf{x}} = \min_{i=1,\cdots,n} \|\mathbf{x} - \mathbf{x}_i\|$, where $\|\cdot\|$ denotes Euclidean distance.

**Proposition 2** (Asymptotic OOD Convergence). *Given non-zero and bounded bandwidth of the Gaussians, then we have almost sure convergence for $\hat{g}_y$ as:*

$$\lim_{d_{\mathbf{x}} \to \infty} \hat{g}_y(\mathbf{x}) = \hat{P}_Y(y).$$

*Proof.* See Appendix A.2 for the proof. $\qquad\square$

## 5 Empirical Results

We conduct several experiments on simulated, OpenML-CC18 [23] [1] and vision benchmark datasets to gain insights on the finite sample performance of KDF and KDN. The details of the simulation datasets and hyperparameters used for all the experiments are provided in Appendix C. For Trunk simulation dataset, we follow the simulation setup proposed by Trunk [24] which was designed to demonstrate 'curse of dimensionality'. In the Trunk simulation, a binary class dataset is used where each class is sampled from a Gaussian distribution with higher dimensions having increasingly less discriminative information. We use both Euclidean and Geodesic distance to detect the nearest polytope (see Equation (4)) on simulation datasets and use only Geodesic distance for benchmark datasets. For the simulation setups, we use classification error, Hellinger distance [25, 26] from the true class conditional posteriors and mean max confidence [4] as performance statistics. While measuring in-distribution calibration for the datasets in OpenML-CC18 data suite, we used maximum calibration error as defined by Guo et al. [18] with a fixed bin number of $R = 15$ across all the datasets. Given $n$ OOD samples, we define OOD calibration error (OCE) to measure OOD performance for the benchmark datasets as:

$$\text{OCE} = \frac{1}{n} \sum_{i=1}^{n} \left| \max_{y \in \mathcal{Y}} (\hat{P}_{Y|X}(y|\mathbf{x}_i)) - \max_{y \in \mathcal{Y}} (\hat{P}_Y(y)) \right|. \tag{18}$$

For the tabular and the vision datasets, we have used ID calibration approaches, such as ISOTONIC [15, 16] and SIGMOID [17] regression, as baselines. Additionally, for the vision benchmark dataset, we provide results with OOD calibration approaches such as: ACET [4], ODIN [6], OE (outlier exposure) [10]. For each approach, $70\%$ of the training data was used to fit the model and the rest of the data was used to calibrate the model.

### 5.1 Empirical Study on Tabular Data

#### 5.1.1 Simulation Study

Figure 1 leftmost column shows 10000 training samples with 5000 samples per class sampled within the region $[-1, 1] \times [-1, 1]$ from the six simulation setups described in Appendix C. Therefore, the empty annular region between $[-1, 1] \times [-1, 1]$ and $[-2, 2] \times [-2, 2]$ is the low density or OOD region in Figure 1. Figure 1 quantifies the performance of the algorithms which are visually represented in Appendix Figure 4. KDF and KDN maintain similar classification accuracy to those of their parent algorithms. We measure hellinger distance from the true distribution for increasing training sample size within $[-1, 1] \times [-1, 1]$ region as a statistics for in-distribution calibration. Column 3 and 6 in Figure 1 show KDF and KDN are better at estimating the ID region compared to their parent methods. In all of the simulations, using geodesic distance measure results in better performance compared to those while using Euclidean distance. For measuring OOD performance, we keep the training sample size fixed at 1000 and normalize the training data by the maximum of their $l2$ norm so that the training data is confined within a unit circle. For inference, we sample 1000 inference points uniformly from a circle where the circles have increasing radius and plot mean max posterior for increasing distance from the origin. Therefore, for distance up to 1 we have in-distribution samples and distances farther than 1 can be considered as OOD region. As shown in Column 4 and 7 of Figure 1, mean max confidence for KDF and KDN converge to the maximum of the class priors, i.e., $0.5$ as we go farther away from the training data origin.

Row 6 of Figure 1 shows KDF-Geodesic and KDN-Geodesic scale better with higher dimensions compared to their Euclidean counterpart algorithms respectively.

#### 5.1.2 OpenML-CC18 Data Study

We use OpenML-CC18 data suite for tabular benchmark dataset study. We exclude any dataset which contains categorical features or NaN values [2] and conduct our experiments on 45 datasets with varying dimensions and sample sizes. For the OOD experiments, we follow a similar setup as that of the simulation data. We normalize the training data by their maximum $l_2$ norm and sample 1000

---

[1] https://www.openml.org/s/99

[2] We also excluded the dataset with dataset id 23517 as we could not achieve better than chance accuracy using RF and DN on that dataset.

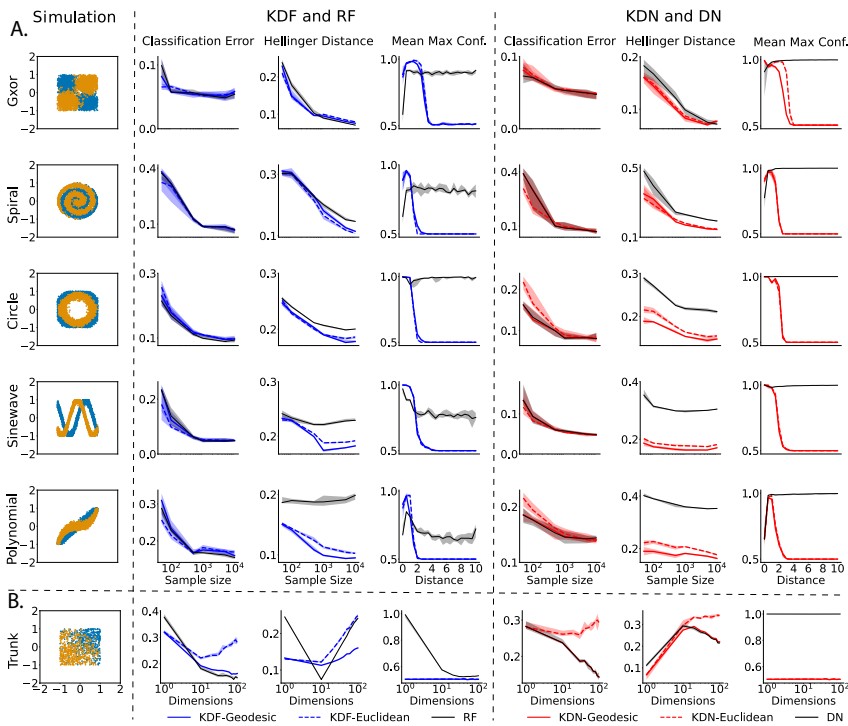

Figure 1: **Simulation datasets, Classification error, Hellinger distance from true posteriors, mean max confidence or posterior for A. five two-dimensional and B. a high dimensional (Trunk) simulation experiments, visualized for the first two dimensions.** The median performance is shown as a dark curve with shaded region as error bars.

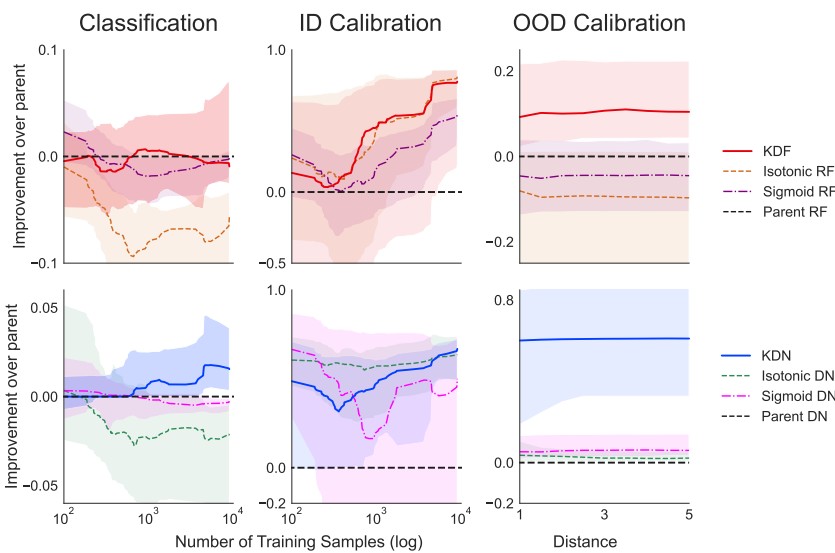

Figure 2: **Performance summary of** KDF **and** KDN **on OpenML-CC18 data suite.** The dark curve in the middle shows the median of performance on 45 datasets with the shaded region as error bar.

testing samples uniformly from hyperspheres where each hypersphere has increasing radius starting from 1 to 5. For each dataset, we measure improvement with respect to the parent algorithm:

$$\frac{\mathcal{E}_p - \mathcal{E}_M}{\mathcal{E}_p}, \tag{19}$$

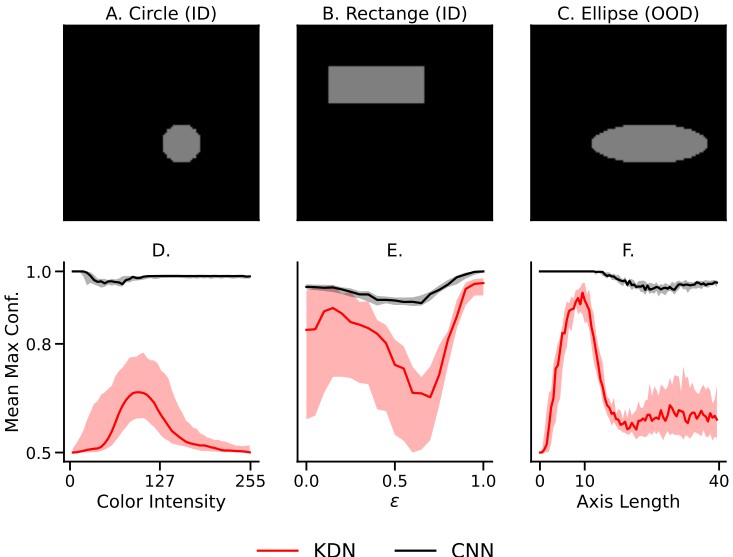

Figure 3: KDN **filters out inference points with different kinds of semantic shifts from the training data.** Simulated images: (A) circle with radius $10$, (B) rectangle with sides $(20, 50)$ and out-of-distribution test points: (C) ellipse with minor and major axis $(10, 30)$. Mean max confidence of KDN are plotted for semantic shift of the inference points created by (D) changing the color intensity, (E) taking convex combination of circle and rectangle, (F) changing one of the axes of the ellipse.

where $\mathcal{E}_p =$classification error, MCE or OCE for the parent algorithm and $\mathcal{E}_M$ represents the performance of the approach in consideration. Note that positive improvement implies the corresponding approach performs better than the parent approach. We report the median of improvement on different datasets along with the error bar in Figure 2. The extended results for each dataset is shown separately in the appendix. Figure 2 left column shows on average KDF and KDN has nearly similar or better classification accuracy compared to their respective parent algorithm whereas ISOTONIC and SIGMOID regression have lower classification accuracy most of the cases. However, according to Figure 2 middle column, KDF and KDN have similar in-distribution calibration performance to the other baseline approaches. Most interestingly, Figure 2 right column shows that KDN and KDF improves OOD calibration of their respective parent algorithms by a huge margin while the baseline approaches completely fails to address the OOD calibration problem.

## 5.2 Empirical Study on Vision Data

In vision data, each image pixel contains local information about the neighboring pixels. To extract the local information, we use convolutional or vision transformer encoders at the front-end. More precisely, we have a front-end encoder, $h_e : \mathbb{R}^D \mapsto \mathbb{R}^m$ and typically, $m << D$. After the encoder there is a few fully connected dense layers for discriminating among the $K$ class labels, $h_f : \mathbb{R}^m \mapsto \mathbb{R}^K$. Note that the $m$-dimensional embedding outputs from the encoder are partitioned into polytopes by the dense layers (see Equation (3)) and we fit a KDN on the embedding outputs. The above approach results in extraction of better inductive bias by KDN from the parent model and makes KDN more scalable with larger parent models and training sample size.

### 5.2.1 Simulation Study

For the simulation study, we use a simple CNN with one convolutional layer (3 channels with $3 \times 3$ kernel) followed by two fully connected layers with 10 and 2 nodes in each. We train the CNN on 2000 circle (radius 10) and 2000 rectangle (sides 20, 50) images with their RGB values being fixed at $[127, 127, 127]$ and their centers randomly sampled within a square with sides 100. The other pixels in the background where there is no object (circle, rectangle or ellipse) were set to 0.

We perform three experiments while inducing semantic shifts in the inference points as shown in Figure 3. In the first experiment, we randomly sampled data similar to the training points. However, we added the same shift to all the RGB values of an inference point (shown as color intensity in Figure 3 D). Therefore, the inference point is ID for color intensity at 127 and otherwise OOD. In the second experiment, we kept the RGB values fixed at $[127, 127, 127]$ while taking convex combination of a circle and a rectangle. Let images of circles and rectangles be denoted by $X_c$ and $X_r$. We derive an interference point as $X_{inf}$:

$$X_{inf} = \epsilon X_c + (1 - \epsilon) X_r \tag{20}$$

Therefore, $X_{inf}$ is maximally distant from the training points for $\epsilon = 0.5$ and closest to the ID points at $\epsilon = \{0, 1\}$. In the third experiment, we sampled ellipse images with the same RGB values as the training points. However, this time we gradually change one of the ellipse axes from $0.01$ to $40$ while keeping the other axis fixed at $10$. As a result, the inference point becomes ID for the axis length of $10$. As shown in Figure 3 (D, E, F), in all the experiments KDN becomes less confident for the OOD points while the parent CNN remains overconfident throughout the semantic shifts of the test points.

### 5.2.2 Vision Benchmark Datasets Study

In this study, we use a $ViT\_B16$ (provided in keras-vit package) vision transformer encoder [27] pretrained on ImageNet [28] dataset and finetuned on CIFAR-10 [29]. We use the same encoder for all the baseline algorithms and finetune it with the corresponding loss function without freezing any weight. As shown in Table 1, pretrained vision transformers are already well-calibrated for ID and the OOD approaches (ACET, ODIN, OE) degrade ID calibration of the parent model. On the contrary, ID calibration approaches (ISOTONIC, SIGMOID) perform poorly compared to that of KDN in the OOD region. KDN achieves a compromise between ID and OOD performance while having reduced confidence on wrongly classified ID samples. The number of populated polytopes (and Gaussians) for KDN is $9323 \pm 353$. See Appendix F for the corresponding experiments using RESNET-50.

Table 1: **KDN achieves good calibration at both ID and OOD regions whereas other approaches which excel either in the ID or the OOD region. Notably,** KDN **has reduced confidence on wrongly classified ID points.** '↑' and '↓' indicate whether higher and lower values are better, respectively. MMC* = Mean Max Confidence on wrongly classified ID points.

|  | Dataset | Statistics | Parent | KDN | ISOTONIC | SIGMOID | ACET | ODIN | OE |
|---|---|---|---|---|---|---|---|---|---|
| ID | CIFAR-10 | Accuracy(%) ↑ | $98.06 \pm 0.00$ | $97.45 \pm 0.00$ | $98.16 \pm 0.00$ | $98.10 \pm 0.00$ | $\mathbf{98.23} \pm 0.00$ | $97.97 \pm 0.00$ | $97.94 \pm 0.00$ |
|  |  | MCE ↓ | $\mathbf{0.00} \pm 0.00$ | $\mathbf{0.00} \pm 0.00$ | $\mathbf{0.00} \pm 0.00$ | $\mathbf{0.00} \pm 0.00$ | $0.01 \pm 0.00$ | $0.02 \pm 0.00$ | $0.01 \pm 0.00$ |
|  |  | MMC* ↓ | $0.76 \pm 0.01$ | $\mathbf{0.65} \pm 0.08$ | $0.74 \pm 0.02$ | $0.90 \pm 0.01$ | $0.86 \pm 0.02$ | $0.97 \pm 0.01$ | $0.69 \pm 0.01$ |
| OOD | CIFAR-100 | OCE ↓ | $0.47 \pm 0.01$ | $\mathbf{0.12} \pm 0.01$ | $0.47 \pm 0.01$ | $0.69 \pm 0.01$ | $0.57 \pm 0.01$ | $0.79 \pm 0.00$ | $0.29 \pm 0.01$ |
|  | SVHN | OCE ↓ | $0.44 \pm 0.06$ | $\mathbf{0.08} \pm 0.02$ | $0.34 \pm 0.12$ | $0.64 \pm 0.16$ | $0.47 \pm 0.04$ | $0.75 \pm 0.03$ | $0.11 \pm 0.02$ |
|  | Noise | OCE ↓ | $0.28 \pm 0.08$ | $0.03 \pm 0.02$ | $0.30 \pm 0.04$ | $0.56 \pm 0.12$ | $\mathbf{0.01} \pm 0.00$ | $0.53 \pm 0.09$ | $0.07 \pm 0.02$ |

## 6   Limitations

Training time complexity for KDF and KDN is $O(n^2 l_f)$ which is dominated by the Geodesic distance calculation. Here $l_f = $ total number of leaves in the forest or total nodes in the dense layers of the network and $n = $ total training samples. However, the distance calculation can be done in parallel using our provided code. Additionally, note that the number of Gaussian kernel used by KDN is upper bounded by number of training samples. Therefore, KDN may not scale for really big datasets like ImageNet [28]. However, the scaling issue may be solved by selectively pruning neighboring polytopes which we will pursue in future.

## 7   Discussion

In this paper, we demonstrated a simple intuition that renders traditional deep discriminative models into a type of binning and kerneling approach. The bin boundaries are determined by the internal structure learned by the parent approach and Geodesic distance encodes the low dimensional structure learned by the model. Moreover, Geodesic distance introduced in this paper may have broader impact on understanding the internal structure of the deep discriminative models which we will pursue in future. Our code, including the package and the experiments in this manuscript, will be made publicly available upon acceptance of the paper.

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

## A  Proofs

### A.1  Proof of Proposition 1

For proving that $d$ is a valid distance metric for $\mathcal{P}_n$, we need to prove the following four statements:

1. $d(r, s) = 0$ when $r = s$.
   **Proof:** By definition, $\mathcal{K}(r, s) = 1$ and $d(r, s) = 0$ when $r = s$.

2. $d(r, s) > 0$ when $r \neq s$.
   **Proof:** By definition, $0 \leq \mathcal{K}(r, s) < 1$ and $d(r, s) > 0$ for $r \neq s$.

3. $d$ is symmetric, i.e., $d(r, s) = d(s, r)$.
   **Proof:** By definition, $\mathcal{K}(r, s) = \mathcal{K}(s, r)$ which implies $d(r, s) = d(s, r)$.

4. $d$ follows the triangle inequality, i.e., for any three polytopes $Q_r, Q_s, Q_t \in \mathcal{P}_n$: $d(r, t) \leq d(r, s) + d(s, t)$.
   **Proof:** Let $\mathcal{A}_r$ denote the set of activation modes in a RELU-net and the set of leaf nodes in a random forest for a particular polytope $r$. $N$ is the total number of possible activation paths in a RELU-net or total trees in a random forest. Below $c(\cdot)$ denotes the cardinality of the set. We can write:

$$N \geq c((\mathcal{A}_r \cap \mathcal{A}_s) \cup (\mathcal{A}_s \cap \mathcal{A}_t)) \tag{21}$$
$$= c(\mathcal{A}_r \cap \mathcal{A}_s) + c(\mathcal{A}_s \cap \mathcal{A}_t) - c(\mathcal{A}_r \cap \mathcal{A}_s \cap \mathcal{A}_t)$$
$$\geq c(\mathcal{A}_r \cap \mathcal{A}_s) + c(\mathcal{A}_s \cap \mathcal{A}_t) - c(\mathcal{A}_r \cap \mathcal{A}_t).$$

Rearranging the above equation, we get:

$$N - c(\mathcal{A}_r \cap \mathcal{A}_t) \leq N - c(\mathcal{A}_r \cap \mathcal{A}_s) + N - c(\mathcal{A}_s \cap \mathcal{A}_t)$$
$$\implies 1 - \frac{c(\mathcal{A}_r \cap \mathcal{A}_t)}{N} \leq 1 - \frac{c(\mathcal{A}_r \cap \mathcal{A}_s)}{N} + 1$$
$$- \frac{c(\mathcal{A}_s \cap \mathcal{A}_t)}{N}$$
$$\implies d(r, t) \leq d(r, s) + d(s, t). \tag{22}$$

### A.2  Proof of Proposition 2

Note that first we find the nearest polytope to the inference point $x$ using Geodesic distance and use Gaussian kernel locally for $x$ within that polytope. Here the Gaussian kernel uses Euclidean distance from the kernel center to $x$ (within the numerator of the exponent). The value out of the Gaussian kernel decays exponentially with the increasing distance of the inference point from the kernel center. We first expand $\hat{g}_y(\mathbf{x})$:

$$\hat{g}_y(\mathbf{x}) = \frac{\hat{f}_y(\mathbf{x})\hat{P}_Y(y)}{\sum_{k=1}^{K} \hat{f}_k(x)\hat{P}_Y(k)}$$
$$= \frac{\tilde{f}_y(\mathbf{x})\hat{P}_Y(y) + \frac{b}{\log(n)}\hat{P}_Y(y)}{\sum_{k=1}^{K}(\hat{f}_k(\mathbf{x})\hat{P}_Y(k) + \frac{b}{\log(n)}\hat{P}_Y(k))}$$

As the inference point $\mathbf{x}$ becomes more distant from training samples (and more distant from all of the Gaussian centers), we have that $\mathcal{G}(\mathbf{x}, \hat{\mu}_r, \hat{\Sigma}_r)$ becomes smaller. Thus, $\forall y$, $\tilde{f}_y(\mathbf{x})$ shrinks. More formally, $\forall y$,

$$\lim_{d_{\mathbf{x}} \to \infty} \tilde{f}_y(\mathbf{x}) = 0.$$

We can use this result to then examine the limiting behavior of our posteriors as the inference point $\mathbf{x}$ becomes more distant from the training data:

$$\lim_{d_\mathbf{x}\to\infty} \hat{g}_y(\mathbf{x}) = \lim_{d_\mathbf{x}\to\infty} \frac{\tilde{f}_y(\mathbf{x})\hat{P}_Y(y) + \frac{b}{\log(n)}\hat{P}_Y(y)}{\sum_{k=1}^K (\tilde{f}_k(\mathbf{x})\hat{P}_Y(k) + \frac{b}{\log(n)}\hat{P}_Y(k))}$$

$$= \frac{(\lim_{d_\mathbf{x}\to\infty} \tilde{f}_y(\mathbf{x}))\hat{P}_Y(y) + \frac{b}{\log(n)}\hat{P}_Y(y)}{\sum_{k=1}^K (\lim_{d_\mathbf{x}\to\infty} \tilde{f}_k(\mathbf{x}))\hat{P}_Y(k) + \frac{b}{\log(n)}\hat{P}_Y(k))}$$

$$= \frac{\hat{P}_Y(y)}{\sum_{k=1}^K \hat{P}_Y(k)}$$

$$= \hat{P}_Y(y).$$

# B   Hardware and Software Configurations

- Operating System: Linux (ubuntu 20.04), macOS (Ventura 13.2.1)
- VM Size: Azure Standard D96as v4 (96 vcpus, 384 GiB memory)
- GPU: Apple M1 Max
- Software: Python 3.8, scikit-learn $\geq$ 0.22.0, tensorflow-macos$\leq$2.9, tensorflow-metal $\leq$ 0.5.0.

# C   Simulations

We construct six types of binary class simulations:

- *Gaussian XOR* is a two-class classification problem with equal class priors. Conditioned on being in class 0, a sample is drawn from a mixture of two Gaussians with means $\pm[0.5, -0.5]^\top$ and standard deviations of 0.25. Conditioned on being in class 1, a sample is drawn from a mixture of two Gaussians with means $\pm[0.5, -0.5]^\top$ and standard deviations of 0.25.

- *Spiral* is a two-class classification problem with the following data distributions: let $K$ be the number of classes and $S \sim \text{multinomial}(\frac{1}{K}\vec{1}_K, n)$. Conditioned on $S$, each feature vector is parameterized by two variables, the radius $r$ and an angle $\theta$. For each sample, $r$ is sampled uniformly in $[0, 1]$. Conditioned on a particular class, the angles are evenly spaced between $\frac{4\pi(k-1)t_K}{K}$ and $\frac{4\pi(k)t_K}{K}$, where $t_K$ controls the number of turns in the spiral. To inject noise along the spirals, we add Gaussian noise to the evenly spaced angles $\theta' : \theta = \theta' + \mathcal{N}(0, 0.09)$. The observed feature vector is then $(r \cos(\theta), r \sin(\theta))$.

- *Circle* is a two-class classification problem with equal class priors. Conditioned on being in class 0, a sample is drawn from a circle centered at $(0, 0)$ with a radius of $r = 0.75$. Conditioned on being in class 1, a sample is drawn from a circle centered at $(0, 0)$ with a radius of $r = 1$, which is cut off by the region bounds. To inject noise along the circles, we add Gaussian noise to the circle radii $r' : r = r' + \mathcal{N}(0, 0.01)$.

- *Sinewave* is a two-class classification problem based on sine waves. Conditioned on being in class 0, a sample is drawn from the distribution $y = \cos(\pi x)$. Conditioned on being in class 1, a sample is drawn from the distribution $y = \sin(\pi x)$. We inject Gaussian noise to the sine wave heights $y' : y = y' + \mathcal{N}(0, 0.01)$.

- *Polynomial* is a two-class classification problem with the following data distributions: $y = x^a$. Conditioned on being in class 0, a sample is drawn from the distribution $y = x^1$. Conditioned on being in class 1, a sample is drawn from the distribution $y = x^3$. Gaussian noise is added to variables $y' : y = y' + \mathcal{N}(0, 0.01)$.

- *Trunk* is a two-class classification problem with gradually increasing dimension and equal class priors. The class conditional probabilities are Gaussian:

$$P(X|Y = 0) = \mathcal{G}(\mu_1, I),$$
$$P(X|Y = 1) = \mathcal{G}(\mu_2, I),$$

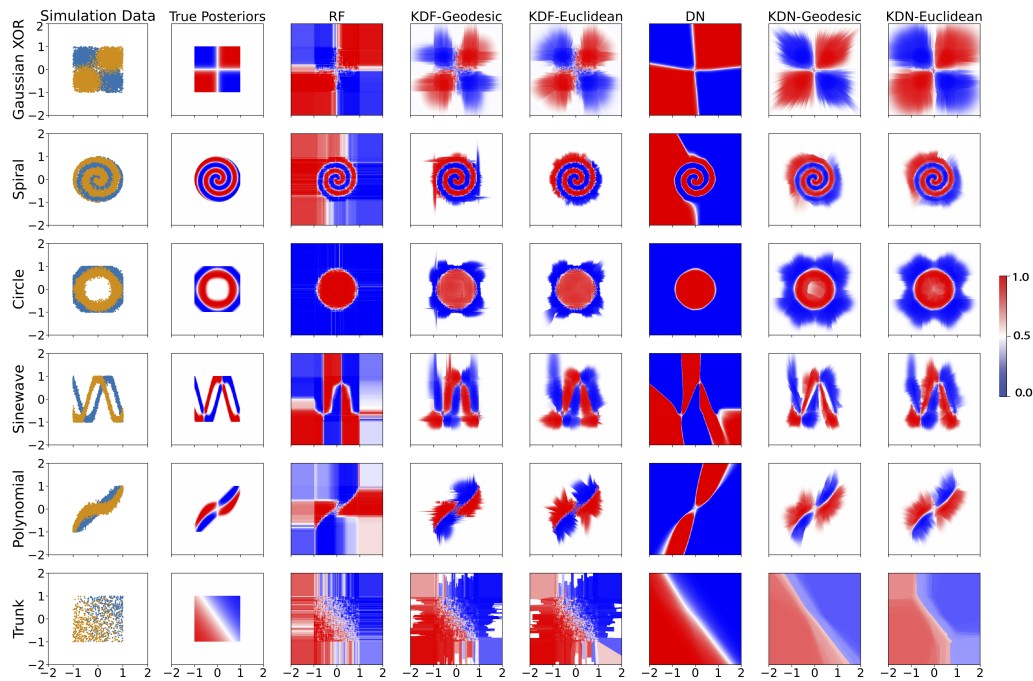

Figure 4: **Visualization of true and estimated posteriors for class 0 from five binary class simulation experiments.** *Column 1*: 10,000 training points with 5,000 samples per class sampled from 6 different simulation setups for binary class classification. Trunk simulation is shown for two dimensional case. The class labels are indicated by yellow and blue colors. *Column 2-8*: True and estimated class conditional posteriors from different approaches. The posteriors estimated from KDN and KDF are better calibrated for both in- and out-of-distribution regions compared to those of their parent algorithms.

where $\mu_1 = \mu, \mu_2 = -\mu$, $\mu$ is a $d$ dimensional vector whose $i$-th component is $(\frac{1}{i})^{1/2}$ and $I$ is $d$ dimensional identity matrix.

Table 2: Hyperparameters for RF and KDF.

| Hyperparameters | Value |
|---|---|
| n_estimators | 500 |
| max_depth | $\infty$ |
| min_samples_leaf | 1 |
| $\lambda$ | $1 \times 10^{-6}$ |
| $b$ | $\exp(-10^{-7})$ |

## D  Pseudocodes

We provide the pseudocode for our porposed algorithms in Algorithm 1, 2 and 3.

## E  Extended Results on OpenML-CC18 data suite

See Figure 5, 6, 7 and 8 for extended results on OpenML-CC18 data suite.

---

**Algorithm 1** Fit a KDX model.

---

**Input:**
    (1) $\theta$          $\triangleright$ Parent learner (random forest or deep network model)
    (2) $\mathcal{D}_n = (\mathbf{X}, \mathbf{y}) \in \mathbb{R}^{n \times d} \times \{1, \ldots, K\}^n$          $\triangleright$ Training data
**Output:** $\mathcal{G}$          $\triangleright$ a KDX model
 1: **function** KGX.FIT$(\theta, \mathbf{X}, \mathbf{y})$
 2:    **for** $i = 1, \ldots, n$ **do**          $\triangleright$ Iterate over the dataset to calculate the weights
 3:        **for** $j = 1, \ldots, n$ **do**
 4:            $w_{ij} \leftarrow$ COMPUTEWEIGHTS$(\mathbf{x}_i, \mathbf{x}_j, \theta)$
 5:        **end for**
 6:    **end for**
 7:
 8:
 9:    $\{Q_r, \mathbf{w}_{rs}\}_{r=1}^{\tilde{p}} \leftarrow$ GETPOLYTOPES$(\mathbf{w})$    $\triangleright$ Identify the polytopes by clustering the samples with similar weight
10:
11:    **for** $r = 1, \ldots, \tilde{p}$ **do**          $\triangleright$ Iterate over each polytope
12:        $\mathcal{G}.\hat{\mu}_r, \mathcal{G}.\hat{\Sigma}_r, \mathcal{G}.\hat{n}_{ry} \leftarrow$ ESTIMATEPARAMETERS$(\mathbf{X}, y, \{\mathbf{w}_{rs}\}_{s=1}^{\tilde{p}})$ $\triangleright$ Fit Gaussians using MLE
13:    **end for**
14:    **return** $\mathcal{G}$
15: **end function**

---

---

**Algorithm 2** Computing weights in KDF

---

**Input:**
    (1) $\mathbf{x}_i, \mathbf{x}_j \in \mathbb{R}^{1 \times d}$          $\triangleright$ two input samples to be weighted
    (2) $\theta$          $\triangleright$ parent random forest with $T$ trees
**Output:** $w_{ij} \in [0, 1]$          $\triangleright$ compute similarity between $i$ and $j$-th samples.
 1: **function** COMPUTEWEIGHTS$(\mathbf{x}_i, \mathbf{x}_j, \theta)$
 2:    $\mathcal{I}_i \leftarrow$ PUSHDOWNTREES$(\mathbf{x}_i, \theta)$ $\triangleright$ push $\mathbf{x}_i$ down $T$ trees and get the leaf numbers it end up in.
 3:    $\mathcal{I}_j \leftarrow$ PUSHDOWNTREES$(\mathbf{x}_j, \theta)$ $\triangleright$ push $\mathbf{x}_j$ down $T$ trees and get the leaf numbers it end up in.
 4:    $l \leftarrow$ COUNTMATCHES$(\mathcal{I}_i, \mathcal{I}_j)$  $\triangleright$ count the number of times the samples end up in the same leaf
 5:    $w_{ij} \leftarrow \frac{l}{T}$
 6:    **return** $w_{ij}$
 7: **end function**

---

---

**Algorithm 3** Computing weights in KDN

---

**Input:**
    (1) $\mathbf{x}_i, \mathbf{x}_j \in \mathbb{R}^{1 \times d}$          $\triangleright$ two input samples to be weighted
    (2) $\theta$          $\triangleright$ parent deep-net model
**Output:** $w_{ij} \in [0, 1]$          $\triangleright$ compute similarity between $i$ and $j$-th samples.
 1: **function** COMPUTEWEIGHTS$(\mathbf{x}_i, \mathbf{x}_j, \theta)$
 2:    $\mathcal{A}_i \leftarrow$ PUSHDOWNNETWORK$(\mathbf{x}_i, \theta)$          $\triangleright$ get activation modes $\mathcal{A}_i$
 3:    $\mathcal{A}_j \leftarrow$ PUSHDOWNNETWORK$(\mathbf{x}_j, \theta)$          $\triangleright$ get activation modes $\mathcal{A}_j$
 4:    $l \leftarrow$ COUNTMATCHES$(\mathcal{A}_i, \mathcal{A}_j)$    $\triangleright$ count the number of times the two samples activate the activation paths in a similar way
 5:    $w_{ij} \leftarrow \frac{l}{N}$          $\triangleright$ $N$ is the total number of activation paths
 6:    **return** $w_{ij}$
 7: **end function**

---

Table 3: Hyperparameters for RELU-net and KDN on Tabular data.

| Hyperparameters | Value |
|---|---|
| number of hidden layers | 4 |
| nodes per hidden layer | 1000 |
| optimizer | Adam |
| learning rate | $3 \times 10^{-4}$ |
| $\lambda$ | $1 \times 10^{-6}$ |
| $b$ | $\exp{(-10^{-7})}$ |

Table 4: ID approaches (SIGMOID, ISOTONIC) are bad at OOD calibration and OOD approaches (ACET, ODIN, OE) are bad at ID calibration. KDN bridges between both ID and OOD calibration approaches. '↑' and '↓' indicate whether higher and lower values are better, respectively. Bolded indicates most performant, or within the margin of error of the most performant.

| | Dataset | Statistics | Parent | KDN | ISOTONIC | SIGMOID | ACET | ODIN | OE |
|---|---|---|---|---|---|---|---|---|---|
| ID | CIFAR-10 | Accuracy(%) ↑ | $77.78 \pm 0.00$ | $76.84 \pm 0.01$ | $\mathbf{78.25} \pm 0.00$ | $76.93 \pm 0.00$ | $75.08 \pm 0.03$ | $78.00 \pm 0.00$ | $73.95 \pm 0.00$ |
| | | MCE ↓ | $0.09 \pm 0.00$ | $\mathbf{0.04} \pm 0.00$ | $\mathbf{0.03} \pm 0.01$ | $0.10 \pm 0.01$ | $0.13 \pm 0.00$ | $0.09 \pm 0.00$ | $0.55 \pm 0.00$ |
| | | MMC* ↓ | $0.47 \pm 0.00$ | $0.37 \pm 0.01$ | $0.54 \pm 0.01$ | $0.43 \pm 0.01$ | $0.69 \pm 0.00$ | $0.48 \pm 0.01$ | $\mathbf{0.13} \pm 0.00$ |
| OOD | CIFAR-100 | OCE ↓ | $0.30 \pm 0.00$ | $0.20 \pm 0.01$ | $0.37 \pm 0.01$ | $0.29 \pm 0.01$ | $0.55 \pm 0.00$ | $0.31 \pm 0.00$ | $\mathbf{0.01} \pm 0.00$ |
| | SVHN | OCE ↓ | $0.87 \pm 0.00$ | $\mathbf{0.01} \pm 0.00$ | $0.85 \pm 0.00$ | $0.69 \pm 0.01$ | $0.90 \pm 0.00$ | $0.87 \pm 0.00$ | $0.04 \pm 0.01$ |
| | Noise | OCE ↓ | $0.90 \pm 0.00$ | $\mathbf{0.00} \pm 0.00$ | $0.87 \pm 0.00$ | $0.71 \pm 0.00$ | $0.01 \pm 0.01$ | $0.06 \pm 0.00$ | $\mathbf{0.00} \pm 0.00$ |

## F    Extended Results on Vision datasets using RESNET-50

In this experiments, we use a RESNET-50 encoder pretrained using contrastive loss [30] as described in http://keras.io/examples/vision/supervised-contrastive-learning. The encoder projects the input images down to a 256 dimensional latent space and we add two dense layers with 200 and 10 nodes on top of the encoder. We use the same pretrained encoder for all the baseline algorithms.

As shown in Table 4, KDN achieves good calibration for both ID and OOD datasets whereas the ID calibration approaches are poorly calibrated in the OOD regions and the OOD approaches have poor ID calibration.

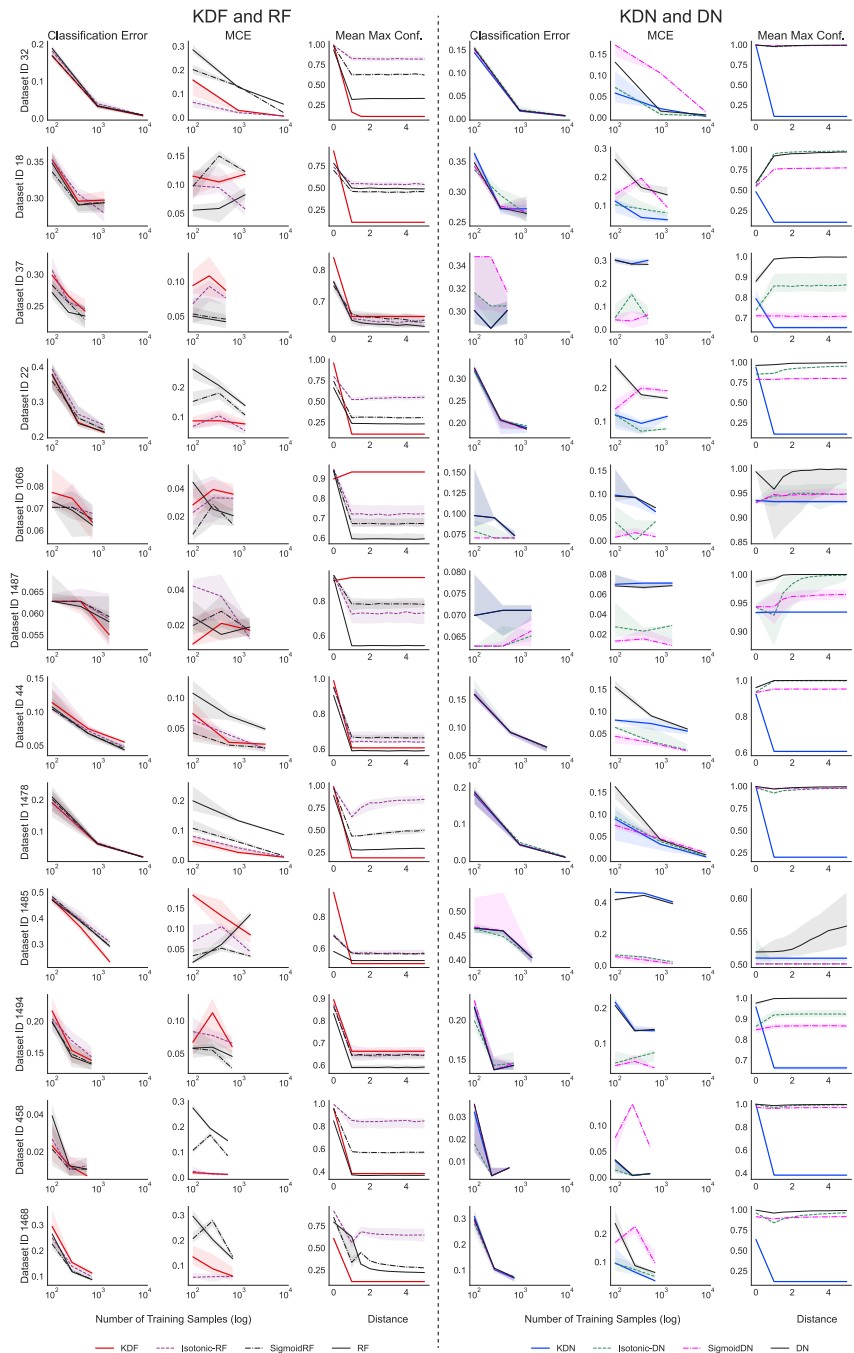

Figure 5: **Extended results on OpenML-CC18 datasets.** *Left:* Performance (classification error, MCE and mean max confidence) of KDF on different Openml-CC18 datasets. *Right:* Performance (classification error, MCE and mean max confidence) of KDN on different Openml-CC18 datasets.

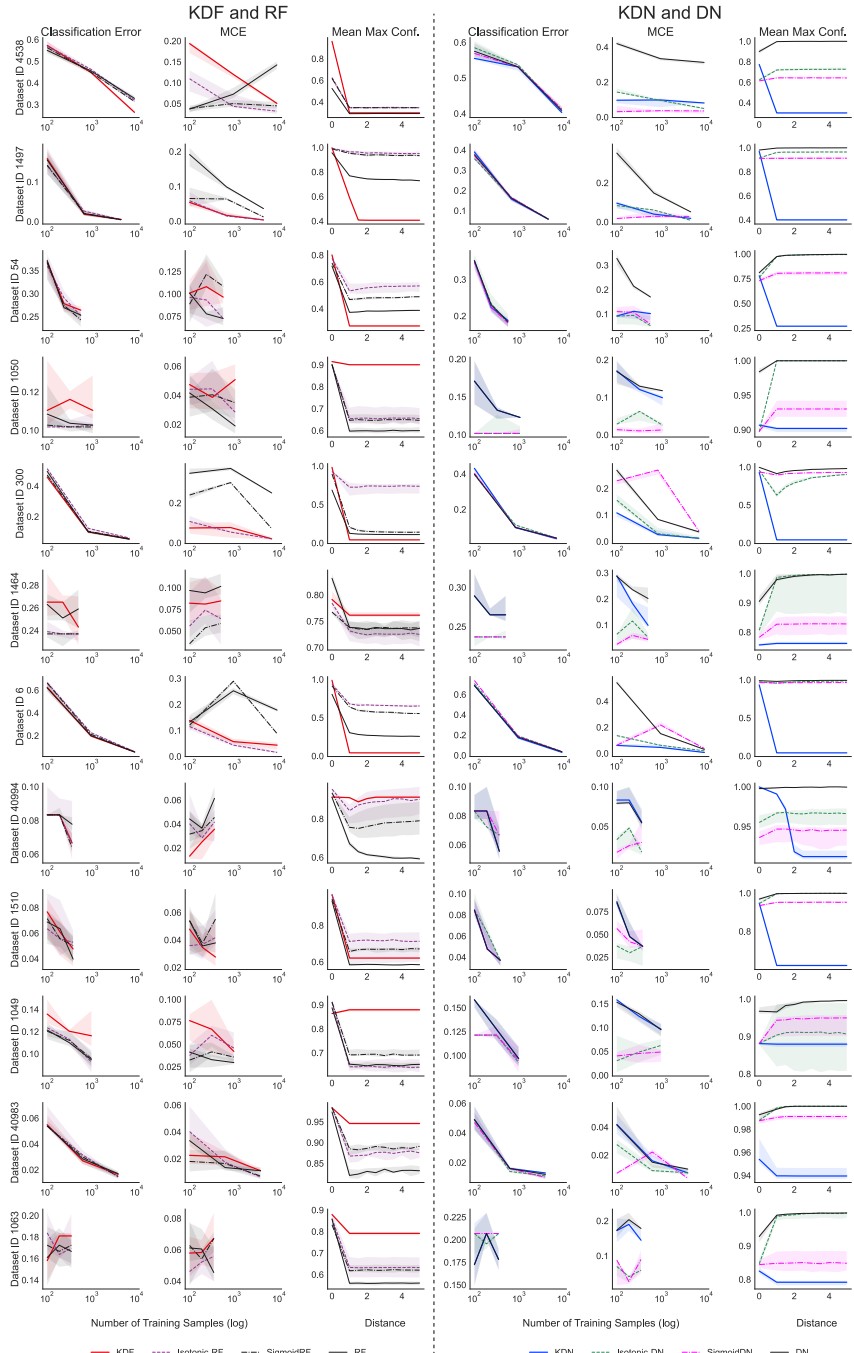

Figure 6: **Extended results on OpenML-CC18 datasets (continued).** *Left:* Performance (classification error, MCE and mean max confidence) of KDF on different Openml-CC18 datasets. *Right:* Performance (classification error, MCE and mean max confidence) of KDN on different Openml-CC18 datasets.

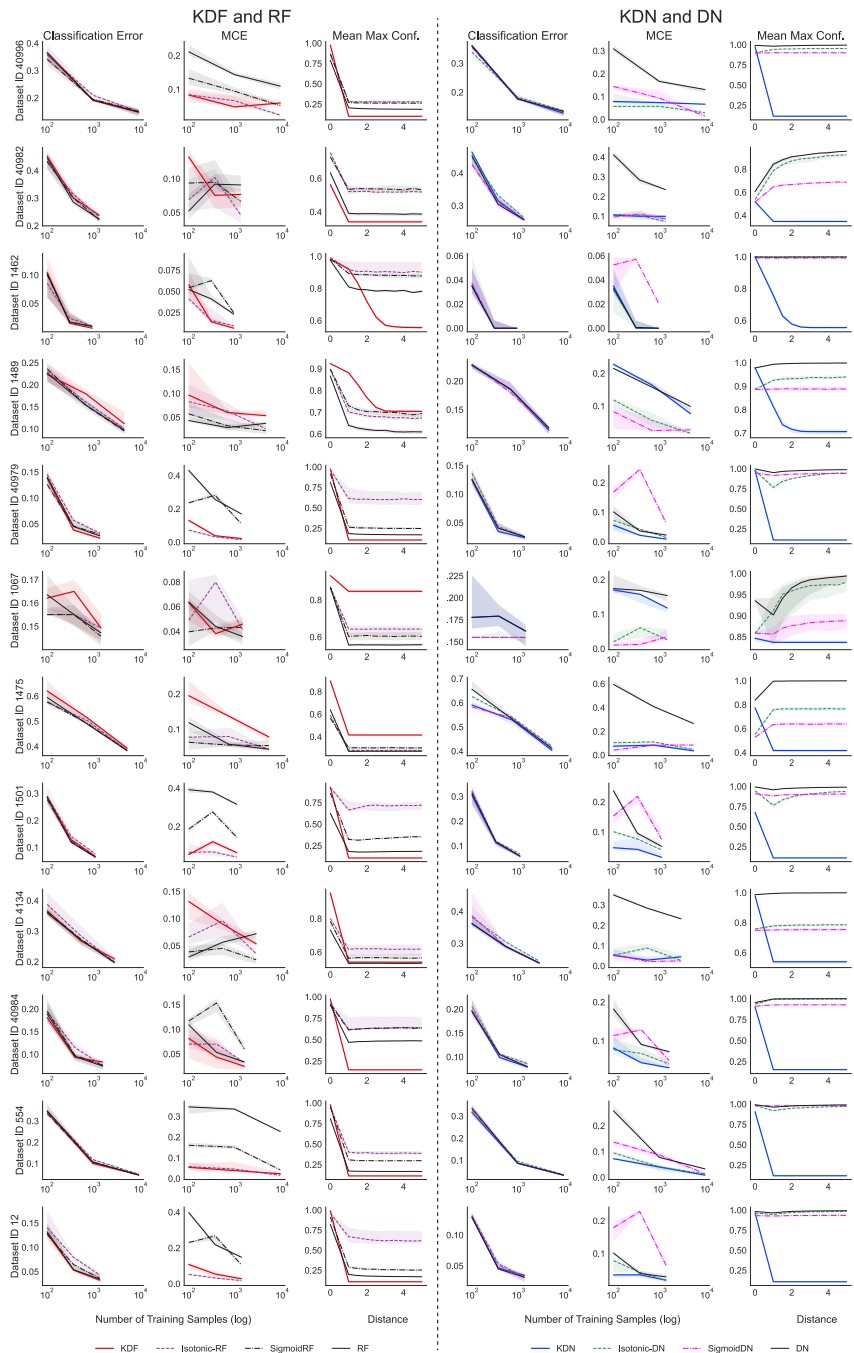

Figure 7: **Extended results on OpenML-CC18 datasets (continued).** *Left:* Performance (classification error, MCE and mean max confidence) of KDF on different Openml-CC18 datasets. *Right:* Performance (classification error, MCE and mean max confidence) of KDN on different Openml-CC18 datasets.

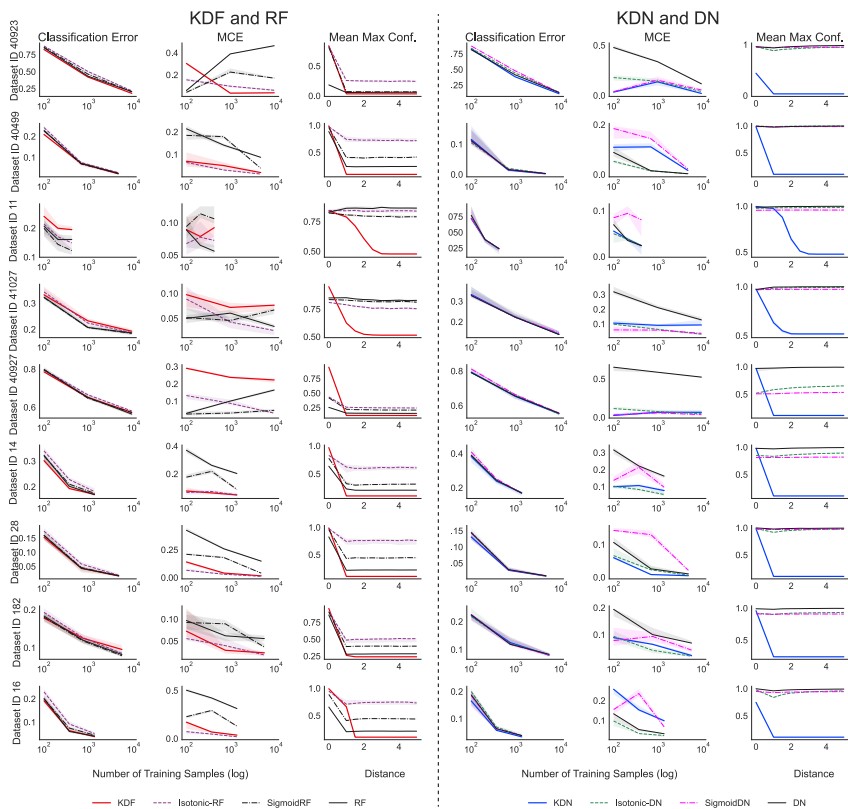

Figure 8: **Extended results on OpenML-CC18 datasets (continued).** *Left:* Performance (classification error, MCE and mean max confidence) of KDF on different Openml-CC18 datasets. *Right:* Performance (classification error, MCE and mean max confidence) of KDN on different Openml-CC18 datasets.

