# OpenReview forum: "Deep Discriminative to Kernel Density Graph for In- and Out-of-distribution Calibrated Inference"
_NeurIPS.cc/2024/Conference — Submitted to NeurIPS 2024_

### Official Review · Reviewer_fbQU · 2024-07-09

**Soundness:** 2
**Presentation:** 3
**Contribution:** 2
**Rating:** 5
**Confidence:** 3

**Summary:**

This paper proposes new methods, Kernel Density Forest (KDF) and Kernel Density Network (KDN), to address issues in confidence calibration for traditional deep learning models and random forests. The motivation stems from the existing literature that deep neural networks using ReLU tend to exhibit high confidence on out-of-distribution (OOD) data due to affine transformations. The proposed methods improve confidence calibration for both in-distribution (ID) and OOD data by partitioning the feature space into polytopes and replacing affine functions within each polytope with Gaussian kernels. Experimental results demonstrate that the proposed methods outperform existing techniques in terms of calibration performance.

**Strengths:**

Originality:
The approach of replacing affine functions within polytopes with Gaussian kernels is novel. The proposed methods address the confidence calibration problem for both ID and OOD data simultaneously, providing an integrated solution to these calibration issues.

Quality:
The theoretical proofs are robust, and the effectiveness of the proposed methods is validated through both simulations and real-world datasets.

Clarity:
The paper is written clearly and concisely.

**Weaknesses:**

Validity of Metrics:
The paper evaluates calibration using Maximum Calibration Error (MCE) for ID data, but does not justify the use of MCE over Expected Calibration Error (ECE) or Adaptive Calibration Error (ACE)[1]. A more detailed explanation and comparison of these metrics would enhance the paper's credibility. Additionally, the definition and justification for OCE (Out-of-distribution Calibration Error) would benefit from a similar comparison with ACE.

[1] https://arxiv.org/abs/1904.01685

Experiments:
To emphasize the effectiveness of the proposed methods, a comparison of execution times would be beneficial, especially since practical applications like web Click-Through Rate (CTR) estimation place significant importance on runtime. The paper should clarify what the noise in Table 1 represents. It would also be advantageous to include experiments on larger and more varied datasets, as well as an evaluation of the methods' performance when combined with in-training calibration methods, which are commonly used alongside post-hoc calibration methods.

**Questions:**

Does the formulation of OCE assume a lack of usable features from the in-distribution domain? What practical and theoretical conditions are required for this assumption? For instance, it is known that large parameter models can improve OOD ECE even when trained with ERM on in-distribution features [2].

[2] https://arxiv.org/abs/2307.08187

**Limitations:**

This paper mentions computational complexity and limitations in practical applications, but lacks detailed experimental results to support these claims. Including such data would provide valuable insights for future research and implementation.

---

> ### Author Rebuttal · Authors · 2024-08-07
>
> Thank you for your thoughtful comments. We are pleased that you recognize the efficacy of our proposed approach in providing an integrated solution for both ID and OOD calibration problems in traditional deep learning models and random forests. We believe we have addressed all your concerns in our responses below. If you find these responses satisfactory, we would greatly appreciate it if you could consider updating your score.
>
> - The paper evaluates calibration using Maximum Calibration Error (MCE) for ID data, but does not justify the use of MCE over Expected Calibration Error (ECE) or Adaptive Calibration Error (ACE)...
> > Thanks for the reference and feedback. We noticed we actually evaluated ECE and called it MCE (please see kdg_code/kdg/utils.py line 36 in the provided codes and definition of ECE in Section 2.1 in [1]). We apologize for the mistake and have corrected it throughout the draft. Additionally, we have also reported ACE in the attached pdf. ECE and ACE provide nearly similar results when the number of classes is low. We noticed ACE gives better estimation of calibration compared to that of ECE when the number of classes is large (for example, cifar100 new results) [1]. This is due to the fact that ECE considers the calibration error only for the predicted class, whereas ACE considers all the classes. Our 45 datasets from OpenML CC-18 suites have (<=) 10 classes [4]. However, we will add the ACE curves for the Openml datasets in the appendix which is similar to the current ECE curves (because of having <=10 classes). We will add the above discussion to the paper and cite the provided reference.
>
> - Additionally, the definition and justification for OCE (Out-of-distribution Calibration Error) would benefit from a similar comparison with ACE.
> > OCE measures OOD calibration and assumes that true class conditional priors for the datasets are known. On the contrary, ACE is used for measuring ID calibration and is a surrogate measure used when true posteriors are not known. For example, in Figure 1 where we know the true distribution, we have used Hellinger distance from the true posteriors instead of ACE. Please see the global response for the corrected definition of OCE. We will add the above clarification in the camera ready version.
>
> - To emphasize the effectiveness of the proposed methods, a comparison of execution times would be beneficial, especially since practical applications like web Click-Through Rate (CTR) estimation place significant importance on runtime.
> > Please see the global response.
>
> - The paper should clarify what the noise in Table 1 represents.
> > We sample noise samples of size $32 \times 32 \times 3$ according to a  Uniform distribution with pixel values within range [0,1]. We will add this text in the draft.
>
> - It would also be advantageous to include experiments on larger and more varied datasets…
> > We have added additional vision experiments using cifar100 (100 classes) and SVHN (10 classes and bigger training size) as ID datasets as suggested by reviewer fbQU. We emphasize that cifar10, cifar100 and SVHN are some of the hardest ID and OOD pairs according to various papers [2, 3], and hence they are adopted as the benchmarking datasets by many of the papers in the literature. Note that doing experiments with extremely large datasets like imagenet (14 million images) is computationally and storage-wise expensive using our current implementation within the rebuttal deadline. We will pursue extremely large datasets in future.  Many relevant papers on OOD calibration use only small- and mid-sized datasets  [5, 6, 7]. We acknowledge this limitation in the paper.
>
> - An evaluation of the methods' performance when combined with in-training calibration methods, which are commonly used alongside post-hoc calibration methods.
> > We have used ACET as an in-training approach followed by our approach to do the proposed experiment using CIFAR100 as ID data. Note that ACET improves OOD calibration leaving less room for improvement for KDN, however KDN improves the ID calibration for ACET. ACET+KDN has same ECE and ACE as KDN and CIFAR-10, SVHN, Noise OCE as 0.12, 0.04, 0.04 respectively. In the end, KDN has nearly similar performance with or without in-training approaches. Moreover, ACET adds a significant computational burden to the whole process. The authors in [8] observed a similar phenomenon.  We think this experiment provides important insights about our approach and we will add it to the appendix.
>
> - Does the formulation of OCE assume a lack of usable features from the in-distribution domain? What practical and theoretical conditions are required for this assumption?
> > The formulation of OCE does not assume a lack of usable features from the ID domain. Our goal is defined in Eq. 1 and OCE measures the OOD calibration error, i.e., difference from the maximum of the class conditional priors in the OOD region according to Eq. 1. For calculating OCE, we need to know the true priors.
>
> - This paper mentions computational complexity and limitations in practical applications, but lacks detailed experimental results to support these claims. Including such data would provide valuable insights for future research and implementation.
> > Please see the global response.
>
> [1] https://arxiv.org/abs/1904.01685
>
> [2] Nalisnick, Eric, et al. "Do deep generative models know what they don't know?."
>
> [3] Fort, Stanislav. "Exploring the limits of out-of-distribution detection."
>
> [4] Bischl, Bernd, et al. "Openml benchmarking suites." arXiv preprint arXiv:1708.03731 (2017).
>
> [5] Gardner, Josh. "Benchmarking distribution shift in tabular data with tableshift."
>
> [6] Borisov, Vadim, et al. "Deep neural networks and tabular data: A survey."
>
> [7] Ulmer, Dennis. "Trust issues: Uncertainty estimation does not enable reliable ood detection on medical tabular data."
>
> [8] Wang, Deng-Bao. "Rethinking calibration of deep neural networks: Do not be afraid of overconfidence."

---

> > ### Comment · Reviewer_fbQU · 2024-08-12
> > **Official Comment by Reviewer fbQU**
> >
> > I thank the authors for their detailed response. I am satisfied with the answers provided, and I would like to raise my score.

---

> > > ### Author Response · Authors · 2024-08-12
> > > **Thanks!**
> > >
> > > Thank you so much for your effort and valuable feedback! They improved our work by a huge margin.

---

### Official Review · Reviewer_gpCb · 2024-07-12

**Soundness:** 3
**Presentation:** 3
**Contribution:** 2
**Rating:** 5
**Confidence:** 4

**Summary:**

The paper proposes a novel approach for OOD detection by learning a posterior distribution that is calibrated for both ID and OOD individuals. It models the class-wise conditional distribution of features by a gaussian kernel respectively for a set of polytopes that cover the feature space. The tail property of gaussian kernels contribute to both ID and OOD calibration. Empirical evidence shows the power of the proposed algorithm across tabular and vision dataset under both ID and OOD settings.

**Strengths:**

The paper is well motivated from the tradeoff of ID calibration and OOD calibration for current approaches for OOD detection methods. The technique of gaussian kernel has a clear geometric intuitive. Compared to affine functions, the tail property ensures that the posterior distribution converges to the prior of labels when a OOD sample deviates far enough from the training support, as proved in Proposition 2. On the other hand, the interpolation by gaussian kernels between neighboring polytopes contributes to ID calibration.

**Weaknesses:**

The major concern is insufficient discussion over the research context of the paper, which renders it hard to precisely evaluate the contribution. The related work section is short. Section 2 shows that "OOD detection" is the closest area to this paper, but this keyword is totally absent from the introduction, where the research area is named "OOD confidence calibration". What is the relation between OOD detection and OOD confidence calibration?
The introduction also reveals two potential approaches for this area: discriminative and generative methods. There are also two settings: ID and OOD confidence calibration. The readers might expect to review current progress for all those categories in the related work section.

**Questions:**

1. In Eq 3, how to ensure the non-negativity of the class-conditional density? The model of affine function might take negative values.
2. The model of class-conditional density with gaussian kernels, referring to Eq 5, resembles Kernel Density Estimation. Could the author please discuss the relation between KDE and the proposed method?
3. In Fig 1, KDF shows a weaker advantage for low dimensional settings. Could the author please explain why?
4. In Fig 2, why are OOD approaches absent? Fig 2 demonstrates the effectiveness of KDF compared to ID approaches in terms of OOD calibration. The readers could be more interested in the performance of OOD approaches.
5. In Table 1, it seems OOD approaches including ACET and ODIN can't even beat the parent model under OOD settings. Could the author please explain why?

**Limitations:**

The author has addressed limitations of their work in terms of sample complexity.

---

> ### Author Rebuttal · Authors · 2024-08-07
>
> We appreciate the reviewer's thoughtful comments. We are glad to see that you recognize the effectiveness of our approach in balancing ID and OOD calibration. We believe we have addressed all your concerns in our responses below. If you find these satisfactory, we would be very grateful if you could consider updating your score.
>
> - The major concern is insufficient discussion over the research context of the paper, which renders it hard to precisely evaluate the contribution. The related work section is short. Section 2 shows that "OOD detection" is the closest area to this paper, but this keyword is totally absent from the introduction, where the research area is named "OOD confidence calibration". What is the relation between OOD detection and OOD confidence calibration?
> > We apologize for the confusion. Our work addresses both in- and out-of-distribution (ID and OOD) calibration. While traditional ID calibration methods like Isotonic and Sigmoid regression focus on achieving calibrated inference within the ID region, they do not address OOD calibration. Conversely, there is another group of literature which is primarily concerned about detecting OOD points. These approaches such as ACET, OE, and ODIN mainly focus on OOD detection rather than OOD calibration. Calibration is harder than detection, akin to how regression is harder than classification.  To see that calibration is harder than detection, consider the fact that a well-calibrated model can perform detection, but a model capable of detecting OOD points may not be calibrated. OOD detection works as long as there are two  distinguishable score sets for ID and OOD points, whereas calibration aims at estimating the true predictive uncertainty of these points. To our knowledge, only Meinke et al. [1] explicitly addressed OOD calibration (Section 3 Theorem 1 in their paper), but they do not consider ID calibration. Our work treats calibration problems as a continuum between the ID and OOD regions rather than addressing them separately. We will revise Section 2 to reflect this discussion.
>
>
>   [1] Meinke, Alexander, Julian Bitterwolf, and Matthias Hein. "Provably Robust Detection of Out-of-distribution Data (almost) for free." arXiv preprint arXiv:2106.04260 (2021).
>
> - In Eq 3, how to ensure the non-negativity of the class-conditional density? The model of affine function might take negative values.
> > The class-conditional density is non-negative because of ReLU activation (please see Section 2 in [1], Figure 1,2,3,4 in [2] for the details). We will clarify in the camera ready version.
>
>   [1] Hein, Matthias, Maksym Andriushchenko, and Julian Bitterwolf. "Why relu networks yield high-confidence predictions far away from the training data and how to mitigate the problem." Proceedings of the IEEE/CVF conference on computer vision and pattern recognition. 2019.
>
>   [2]  Xu, Haoyin, et al. "When are deep networks really better than decision forests at small sample sizes, and how?." arXiv preprint arXiv:2108.13637 (2021).
>
> - The model of class-conditional density with gaussian kernels, referring to Eq 5, resembles Kernel Density Estimation. Could the author please discuss the relation between KDE and the proposed method?
> > This is an excellent point! We agree there are similarities between KDE and the proposed method. However, there is an indicator function in Eq. 5 which is implemented using the geodesic distance proposed in the draft (which is absent in KDE), which makes KDN, KDF scale better with higher dimensions. Moreover, the center and bandwidth of the Gaussians are estimated in a data-driven manner using the representation learned by the parent discriminative model. We will add the above clarification after Eq 5.
>
> - In Fig 1, KDF shows a weaker advantage for low dimensional settings. Could the author please explain why?
> > In Fig 1, each Random Forest is an ensemble of 500 decision trees. An ensemble of uncalibrated learners has improved calibration over the individual uncalibrated learner [1]. This phenomenon leaves less room for improvement for KDF in low dimensional settings. On the contrary, the deep-net models are standalone learners with poor calibration which can be improved a lot by KDN.
>
>   [1]  Stickland, Asa Cooper, and Iain Murray. "Diverse ensembles improve calibration." arXiv preprint arXiv:2007.04206 (2020).
>
> - In Fig 2, why are OOD approaches absent? Fig 2 demonstrates the effectiveness of KDF compared to ID approaches in terms of OOD calibration. The readers could be more interested in the performance of OOD approaches.
> > ACET, ODIN, OE are tailor-made for vision problems, therefore we can not run them on tabular data using the author provided codes. . To the best of our knowledge, the tabular OOD method [1] that we found does OOD detection, not calibration. As it does not yield any posterior, we could not benchmark with the above method. We will add the above explanations to Section 5.1.2.
>
>     [1] Ren, Jie, et al. "Likelihood ratios for out-of-distribution detection." Advances in neural information processing systems 32 (2019).
>
> - In Table 1, it seems OOD approaches including ACET and ODIN can't even beat the parent model under OOD settings. Could the author please explain why?
> > ACET and ODIN highly depend on the model architecture and the nature of the ID and OOD testsets. Moreover, they also depend on the OOD set used to train them. We used the OOD set used by the authors of the above algorithms. All these factors contribute to their inconsistent performance across datasets and model architecture. The authors of [1] found a qualitatively similar result.
>
>   [1] “Tajwar, Fahim, et al. "No true state-of-the-art? ood detection methods are inconsistent across datasets." arXiv preprint arXiv:2109.05554 (2021).”

---

> > ### Comment · Reviewer_gpCb · 2024-08-08
> >
> > I acknowledge and thank the author for their response. In the rebuttal, the author has addressed the paper’s relation to OOD detection in detail. My remaining concern is about the relation between OOD calibration and IID calibration. It is an important and relevant problem to learn a posterior distribution that is calibrated simultaneously for both ID and OOD samples. However, as noted by other reviewers, it is confusing why a metric like ECE or ACE is not simultaneously adopted for both settings, which could have made the results more convincing.
> >
> > The author has claimed ECE as a metric only for IID calibration, but I do not see a specific distribution assumption for ECE, and I believe ECE is still valid out-of-distribution. To illustrate this, consider the extreme setting where the outcome is independent of the feature. In this case, ECE is 0 if and only if the predictor outputs the prior label distribution, which behaves similarly to OCE. Additionally, ECE has the advantage of measuring calibration error when the distribution shift is moderate, such that the feature is still predictive, albeit less so, for the label.

---

> > > ### Author Response · Authors · 2024-08-08
> > >
> > > Thanks for elaborating the above concern. Sorry that we did not understand it fully previously.  The idea seems intuitive for measuring calibration in transfer learning settings with distribution shift in the feature space for the target task. We are not sure we understand how to measure ECE for OOD points in our setting. According to the definition of ECE, the calibration error is the difference between the fraction of predictions in the bin that are correct (accuracy) and the mean of the probabilities in the bin (confidence).  To calculate ECE, we need to measure accuracy. OOD points are unsupervised in our setting, i.e., there is no label associated with an OOD point. Our goal (Eq. 1 in our paper) is to calibrate the model so that it knows whenever it faces an OOD point (same as existing OOD detection approaches). There is not a target task in this setting where we want to do transfer learning. If a model is well-calibrated in the OOD region, it will always predict the majority (max prior) class in the ID training data for the OOD points. How do we calculate accuracy in this case? To clarify it further, consider we train a model on CIFAR10 and test it on OOD points from CIFAR100. The model will predict class labels within 1 to 10, but CIFAR100 has 100 classes. To the model (if it is calibrated) CIFAR100 should look like unknown points and hence it will have confidence at the prior level.

---

> > > > ### Comment · Reviewer_gpCb · 2024-08-08
> > > >
> > > > Thank the author for their detailed and patient responses. To summarize, the author has explained how OOD calibration is connected to and distinguished from calibration (ID) and OOD detection, respectively. From the rebuttal, it is revealed that OOD calibration is a new setting/metric proposed by this paper, which is certainly a contribution. (Although the author mentioned that a similar idea is only studied in a particular theorem of [1], the literature is also well-established for OOD detection, as is also evidenced by the title.) In my original review, I identified a weakness in the unclear discussion of related work, which was largely due to the mixed usage of OOD detection and OOD calibration. The author has clarified in the rebuttal that ACET, OE, and ODIN, stated in the paper as OOD calibration methods, are actually focused on OOD detection. Given the contribution of setting up OOD calibration, which lies somewhere between OOD detection and calibration, I still have concerns about whether this setting, as well as the introduced method, has a significant edge over both related fields.
> > > >
> > > > **OOD Calibration vs. OOD Detection**: The author claims that OOD calibration is more challenging because it involves predicting a continuous and aligned confidence, while OOD detection is a binary decision. However, the author also mentions that the empirical study is conducted without access to labels, even at the evaluation stage, resulting in a metric OCE that depends completely on a prior of labels. This experimental setting essentially reduces the task to a binary one: whether to output normal confidence for an ID sample or just a prior for an OOD sample. A binary OOD detector can also immediately output a confidence based on the prior if a sample is identified as OOD. Therefore, the proposed method is essentially an OOD detector, which implies the need to reconsider both the theoretical and empirical results within the broader literature of OOD detection.
> > > >
> > > > **OOD Calibration vs. Calibration**: As stated above, OOD calibration defined in an unsupervised manner is closer to OOD detection. However, there is established research for calibration [2] under distribution shift, which studies algorithms to produce calibrated outputs for different distributions. I believe this line of research is also relevant to the new notion.
> > > >
> > > > Overall, I remain concerned about the exact boundary of the newly proposed notion of OOD calibration and the introduced methodology. Therefore, I maintain my evaluation of this paper.
> > > >
> > > > [1] Meinke, Alexander, Julian Bitterwolf, and Matthias Hein. “Provably Robust Detection of Out-of-distribution Data (almost) for free.” arXiv preprint arXiv:2106.04260 (2021).
> > > >
> > > > [2] Kim, M. P., Kern, C., Goldwasser, S., Kreuter, F., & Reingold, O. (2022). Universal adaptability: Target-independent inference that competes with propensity scoring. Proceedings of the National Academy of Sciences, 119(4), e2108097119.

---

> ### Author Response · Authors · 2024-08-09
>
> - “This experimental setting essentially reduces the task to a binary one: whether to output normal confidence for an ID sample or just a prior for an OOD sample. ”
> > The reviewer is correct, in the experimental setting, as one moves further away from the ID setting, an OOD calibrated classifier will output the prior.  However, close to the ID setting, the OOD calibrated classifier will output a probability that the sample is in any given class.  This is in contrast to an OOD detector, which, regardless of how close or far the data are from the ID data, if is OOD, it always effectively outputs the prior.
> - “A binary OOD detector can also immediately output a confidence based on the prior if a sample is identified as OOD.”
> > Yes, a binary OOD detector can output a confidence based on the prior.  However, without calibration, there is no reason to expect that confidence to be….calibrated.  This is precisely why the OOD calibration is more difficult, and more informative, than pure OOD detection.
> - “Therefore, the proposed method is essentially an OOD detector, which implies the need to reconsider both the theoretical and empirical results within the broader literature of OOD detection.”
> > We would say that our method subsumes OOD detection, because it also includes ID calibration, and OOD calibration. To our knowledge, there are no other papers demonstrating any algorithm with all these properties.

---

### Official Review · Reviewer_tAq4 · 2024-07-12

**Soundness:** 3
**Presentation:** 3
**Contribution:** 3
**Rating:** 7
**Confidence:** 3

**Summary:**

The paper introduces a way to calibrate ReLU networks or random forests by breaking them down into piecewise linear functions on polytopes and replacing the linear parts with Gaussian kernels. This approximation allows to naturally calibrate the models for the ID domain, where confidence will be high due to the density of ID samples that translates into high kernel values, and for the OOD domain, where confidence will be low due to the large distance to ID samples.

**Strengths:**

- The method is novel and mathematically grounded
- The presentation is clear
- The benchmarks are OK

**Weaknesses:**

The main weakness I find is about the computational time of the method. The number of polytopes scales exponentially with the number of neurons, so I am concerned with the applicability of the method to large (or even medium-scale) neural networks. What is the computational cost of the method for the considered benchmarks, in terms of runtime?

The toy simulations are unnecessarily tedious to grasp and take up a lot of space. I do not say that they are complex, but they hinder the reading flow and do not bring much to the presentation. I would advise putting some of them in the appendix to leave more space for other explanations. Indeed, Section 5 is difficult to read (many "chunk" paragraphs with mathematical notations) and would benefit from more structured writing and more flow.

**Questions:**

1. Section 4.4 How is $\omega_{rs}$ estimated in that case?
2. Table 1: what is the Parent approach exactly compared to KDN and KDF? The definition of Section 3.3 does not seem to describe a whole model but only the parent idea that is then declined with KDN and KDF.
3. What would be the AUROC, FPR, i.e. metrics commonly used in OOD detection?
4. Figure 1. What is "dimensions" in x-axis?
5. l. 192 I do not get how OCE measures an error since it only includes estimated quantities, and I do not see the estimated confidence score $\hat{g}_y(x)$. In addition, are $x_i$ OOD samples in that case? If yes it should be made explicit.
6. l. 213 I disagree that normalizing ensures that distances up to 1 are ID and above are OOD. There can be "holes" in the distribution (as the circle dataset), or modes. What is the authors' opinion about that?

**Limitations:**

The authors have adequately addressed the limitations

---

> ### Author Rebuttal · Authors · 2024-08-07
>
> We thank the reviewer for the intuitive comments. We are glad that the reviewer recognized the intuition behind our approach . We believe we have addressed all your concerns in the response below. If you think these responses are satisfactory, we would be very grateful if you can update your score.
>
>
> - The main weakness I find is about the computational time of the method. The number of polytopes scales exponentially with the number of neurons, so I am concerned with the applicability of the method to large (or even medium-scale) neural networks. What is the computational cost of the method for the considered benchmarks, in terms of runtime?
> > Please see the global response.
>
> - The toy simulations are unnecessarily tedious to grasp and take up a lot of space. I do not say that they are complex, but they hinder the reading flow and do not bring much to the presentation. I would advise putting some of them in the appendix to leave more space for other explanations. Indeed, Section 5 is difficult to read (many "chunk" paragraphs with mathematical notations) and would benefit from more structured writing and more flow.
> >  This is an excellent suggestion! We will put rows four of the rows of Figure 1 to appendix. We will reorganize the chunk paragraphs and make Section 5 more concise in the camera ready version, and put the formal equations in the appendix as necessary.
>
> - Section 4.4 w_{rs} How is  estimated in that case?
> > We apologize for missing it. We have rewritten line 159-160 as “We estimate $w_{rs}$ by exponentiating the above kernel using Equation 15.“
>
> - Table 1: what is the Parent approach exactly compared to KDN and KDF? The definition of Section 3.3 does not seem to describe a whole model but only the parent idea that is then declined with KDN and KDF.
> > Sorry for the confusion. In Table 1, by “Parent approach” we mean the original vision transformer [1] that was trained on CIFAR10. We will update the table accordingly.
>
>    [1] https://pytorch.org/vision/main/models/generated/torchvision.models.vit_b_16.html
>
> - What would be the AUROC, FPR, i.e. metrics commonly used in OOD detection?
> >  We have added AUROC and FPR in the updated table in the attached pdf. KDN has nearly similar AUROC and FPR to those of OOD detection approaches. Note that these scores are used for OOD detection. However, we are addressing both ID and OOD calibration (Eq. 1 in our paper). OOD detection works as long as there are two  distinguishable score sets for ID and OOD points, whereas calibration aims at estimating the true predictive uncertainty of these points. That being said, a well-calibrated model can perform detection, but a model capable of detecting OOD points may not be calibrated. We will add the above discussion in Section 2.
>
> - Figure 1. What is "dimensions" in x-axis?
> > We will replace it with “Number of dimension” which indicates the number of increasing dimensions from the Trunk simulation.
>
> - l. 192 I do not get how OCE measures an error since it only includes estimated quantities, and I do not see the estimated confidence score
> > Thanks for this intuitive observation and catching the mistake. Please see the global response.
>
> - In addition, are  x_i OOD samples in that case? If yes it should be made explicit.
> > We have rewritten line 192 as: “Given n OOD samples {x_i}_{i=1}^n, we define OOD calibration error (OCE) to measure OOD performance for the benchmark datasets as:”
>
> - l. 213 I disagree that normalizing ensures that distances up to 1 are ID and above are OOD. There can be "holes" in the distribution (as the circle dataset), or modes. What is the authors' opinion about that?
> > This is an excellent catch and we agree! We have rewritten line 213 as: “Therefore, ID samples are confined within distance 1. “

---

> > ### Comment · Reviewer_tAq4 · 2024-08-08
> >
> > I thank the authors for their clarification and raised my rating. I strongly encourage them to polish the presentation for the camera ready version.

---

> > > ### Author Response · Authors · 2024-08-08
> > > **Thanks!**
> > >
> > > Thank you so much for your intuitive and valuable feedback! It improved our work significantly. We will make sure our presentation is significantly improved taking in consideration all the points raised by you for the camera ready version.

---

### Author Rebuttal · Authors · 2024-08-07

We thank all the reviewers for their strenuous effort and time to go through our paper and provide valuable feedback. Below, we address the common concerns:

- Reviewers were concerned about the runtime of our approach, possibly it could be an exponential function of the number of nodes. However, we did additional experiments (see Fig. 10 in the attached pdf) where we show training and testing time both are linear in the number of the nodes. Moreover, training time is only 200 seconds even when there are 40,000 nodes running on a MacBook Pro with an Apple M1 Max chip and 64 GB of RAM. The number of total polytopes in KDN is upper bounded by the training sample size as we only consider the polytopes populated by training data (see the first paragraph of Section 3.3 and Eq. 5). We will add this figure in the camera ready version.

- Reviewers asked about the training time complexity of other baseline approaches. OOD calibration approaches such as ACET, OE and ODIN take about 2 days, an hour, 6 hours, respectively on GPUs. In-distribution calibration methods such as isotonic regression and sigmoid regression take a few minutes and use CPUs. Our approach addresses both ID and OOD calibration while taking a few minutes to train on CPUs, rather than GPUs. All the computations were performed for producing the results in Table 1 using a MacBook Pro with an Apple M1 Max chip and 64 GB of RAM. We will add these numerical results to the camera ready version.


- Reviewers were concerned about the definition of OCE. We have corrected our OCE definition in Eq 18. Previously we erroneously used estimated priors and now we have replaced it with the true priors:

$$\text{OCE} = \frac{1}{n} \sum_{i=1}^n \left|\max_{y \in \mathcal{Y}}(\hat{P}_{Y|X}(y|\mathbf{x}_i)) - \max\_{y\in \mathcal{Y}}(P_Y(y)) \right|.$$

We can only calculate OCE when we know the true priors. For all the experiments, we fixed the priors and sampled our training data accordingly. We have fixed the equation in the paper, and will comment about the requirement to assume the priors are known.

---

### Decision · Program_Chairs · 2024-09-25

**Decision:**

Reject

**Comment:**

**Paper summary**

This paper addresses the problem of learning a probabilistic classifier (or a posterior distribution) that is both in-distribution (ID) and out-of-distribution (OOD) calibrated. The idea starts from the observation that a function modeled by a deep ReLU network can be equivalently written as a union of polytopes with affine activation functions. The proposed idea replaces the affine function with a Gaussian kernel. Relying on the fact that a Gaussian kernel decays exponentially quickly as one moves away from its center, the constructed posterior model can be made to recover the class prior distribution as a test point becomes far away from training examples. This alleviates the problem of model overconfidence when given an OOD test example and helps with OOD calibration.

**Review**

All reviewers note that the proposed method is well motivated. The proposed solution nicely addresses ID and OOD calibration at the same time (fbQU). The method is mathematically grounded (Reviewer tAq4). Runtime of the method was an issue raised by a few reviewers. The rebuttal sufficiently addressed this issue with results showing the relationship of training time and the number of hidden nodes.

Sticking issues discussed during the discussion phase were 1. novelty of the Gaussian-based model, and 2. preciseness of Proposition 2.

1. The Gaussian-kernel-based construction of a posterior distribution that decays to the prior as one moves away from the training points is not entirely new. This property has been used in, for instance, the Gaussian process regression literature (among others). It can be useful for OOD calibration but the novelty is limited.

2. Proposition 2 (arguably the main theorem in the work) is not precise for a few reasons. Firstly it is unclear for what random variable the almost-sure convergence applies to. "Almost sure" convergence is mentioned in the statement. Secondly, its implication on OOD calibration is unclear. Proposition 2 ensures that as one moves away from the training points, the posterior reduces to the prior, ensuring no overconfidence issues. However, this is an asymptotic statement.  As Reviewer gpCb sums up nicely, “for calibration, we expect a non-asymptotic guarantee that ensures the output confidence is accurate”.

Recommendation: reject.